# L-phenylalanine Increased Gut Hormone Secretion through Calcium-Sensing Receptor in the Porcine Duodenum

**DOI:** 10.3390/ani9080476

**Published:** 2019-07-24

**Authors:** Jiangyin Feng, Cuicui Kang, Chao Wang, Liren Ding, Weiyun Zhu, Suqin Hang

**Affiliations:** National Center for International Research on Animal Gut Nutrition, Laboratory of Gastrointestinal Microbiology, College of Animal Science and Technology, Nanjing Agricultural University, Nanjing 210095, China

**Keywords:** L-phenylalanine, calcium-sensing receptor, gut hormone, pig, duodenum

## Abstract

**Simple Summary:**

The proper modulation of feed intake not only meets the nutrient requirement for the maximum growth rate in pigs, but also avoids feed waste. A complete understanding of how dietary factors affect the appetite in pigs will provide a strategy to modulate the feed intake of pigs. L-phenylalanine (L-Phe) has been demonstrated to induce satiety through stimulating anorectic hormone secretion in rodents. However, whether L-Phe has similar effects in pigs is unknown. Here, we aimed to investigate how L-Phe affects gut hormone secretion, along with insight into the underlying mechanism in porcine duodenum by using an in vitro perfusion system. Results showed that 80 mM L-Phe triggered glucose-dependent insulinotropic peptide (GIP) and cholecystokinin (CCK) release, and also upregulated calcium-sensing receptor (CaSR) and its downstream molecules, such as protein kinase C (PKC) and inositol 1,4,5-triphosphate receptor (IP_3_R) expression. However, these effects were attenuated by treatment with a CaSR antagonist. Our findings show that CaSR participates in Phe-induced hormone secretion in pig duodenum, indicating that CaSR may be a potential target in the food intake regulation of pigs.

**Abstract:**

Luminal amino acids have a pivotal role in gut hormone secretion, and thereby modulate food intake and energy metabolism. However, the mechanisms by which amino acids exert this effect remains unknown. The purpose of this research was to investigate the response of L-phenylalanine (L-Phe) to gut hormone secretion and its underlying mechanisms by perfusing the pig duodenum. Eighty mM L-Phe and extracellular Ca^2+^ stimulated cholecystokinin (CCK) and glucose-dependent insulinotropic peptide (GIP) release, and upregulated the mRNA expression of the calcium-sensing receptor (CaSR), CCK, and GIP. Western blotting results showed that L-Phe also elevated the protein levels of CaSR, the inositol 1,4,5-triphosphate receptor (IP_3_R), and protein kinase C (PKC). However, the CaSR inhibitor NPS 2143 reduced the mRNA expression of CaSR, CCK, and GIP, and the secretion of CCK and GIP, as well as the protein level of CaSR, IP_3_R, and PKC. These results indicated that Phe stimulated gut secretion through a CaSR-mediated pathway and its downstream signaling molecules, PKC and IP_3_R.

## 1. Introduction

In commercial pig production, it is critical to meet the maximum growth rate nutrient requirement without wasting feed, emphasizing the importance of understanding how dietary factors can affect the feed intake of pigs [1]. Evidences suggest that a high-protein diet is capable of suppressing food intake and reducing weight gain, and these effects may be associated with the sensing of its degradation products, especially amino acids in the gastrointestinal tract (GIT) [2,3]. Amino acids are an essential ingredient of swine diets, and are extensively generated by protein digestion in the small intestine. Therefore, identifying the mechanisms underlying amino acid-induced satiety will contribute to more effective nutritional manipulation for the modulation of feed intake in porcine production.

Luminal amino acids function as the building blocks of proteins, as well as an essential releaser of anorexigenic gut hormones to enhance satiety [4]. Various types of amino acids exert different effects on gut peptide secretion, due to their specific structural characteristics and physicochemical properties [5]. Among 20 proteinogenic amino acids, aromatic amino acids—mainly L-phenylalanine (L-Phe) and L-tryptophan (L-Trp)—have become subjects of great interest, as some previous research reported their potent secretory effect on rodents’ small intestines for the release of glucagon-like peptide-1 (GLP-1), glucose-dependent insulinotropic peptide (GIP) [6], and CCK [7].

With the detection of G-protein-coupled receptors (GPCRs) expressed in the gut, a lot of research has established their involvement in the regulation of gut hormone secretion induced by luminal nutrients [8,9]. A class C G-protein-coupled receptor, the calcium-sensing receptor (CaSR), was originally considered as the ion sensor used to maintain extracellular Ca^2+^ ([Ca^2+^]_e_) homeostasis [10]. Currently, emerging evidence is postulating that CaSR also functions as a sensor of aromatic amino acids, and thereby mediates gut hormone secretion. Intra-ileal administration of L-Phe has been shown to increase plasma GLP-1 levels and reduce food intake in rats, whereas these effects are blunted by the co-administration of the CaSR inhibitor [11], which supports the suggestion that CaSR is indispensable in Phe-induced GLP-1 secretion. However, the cellular mechanisms by which the recognition of Phe by the CaSR triggers gut hormone secretion remains unclear.

Evidence suggests that the elevation of intracellular calcium ([Ca^2+^]_i_) is a crucial step of CaSR-mediated gut hormone secretion [12,13]. Besides the opening of plasma membrane ion channels which lead to [Ca^2+^]_e_ influx, the activation of the inositol 1, 4, 5-triphosphate receptor (IP_3_R) also contributes to the upregulation of [Ca^2+^]_i_. IP_3_R is a membrane glycoproteins complex, and acts as a Ca^2+^ channel that is mainly located on the endoplasmic reticulum [14]. With the combination of IP_3_R and its specific activator IP_3_, which is generated by hydrolysis of phosphatidylinositol 4,5-bisphosphate (PIP2), the free Ca^2+^ would be instantly released from the endoplasmic reticulum, resulting in Ca^2+^ oscillation [15]. In addition, [Ca^2+^]_i_ is an indispensable regulator for protein kinase C (PKC) activation, which controls the function of other proteins through multiple phosphorylation mechanisms [16]. A previous study shows how the activation of PKC stimulated GLP-1-induced insulin secretion in mouse and human islets [17]. Therefore, considering the significance of [Ca^2+^]_i_ in the regulation of gut hormone release, both IP_3_R and PKC may participate in gut hormone secretion induced by Phe.

While many studies have shown that amino acids induce gut hormone secretion through the activation of CaSR in rodents, it remains unknown whether CaSR exerts similar effects in pigs. Our previous research found that Trp has the potential to stimulate CCK and GIP release from pig duodenal tissue via a CaSR-dependent pathway [18], suggesting porcine CaSR also plays a vital role in mediating gut hormone secretion. Although both Trp and Phe belong to aromatic amino acids, they show different profiles in the involvement of anabolism, even in the recognition of luminal chemosensory receptors. For instance, Daly et al. reported that TAS1R1/TAS1R3, another amino acid sensing receptor, was activated in the presence of Phe in the mouse proximal intestine, but not Trp [7]. Therefore, it is necessary to gain a complete understanding of the relationship between CaSR and aromatic amino acids in pigs.

The aim of this study was to investigate: (1) the effects of Phe on CCK and GIP secretion in the duodenum of swine by using a perfusion system; (2) the role of CaSR in mediating these effects; and (3) the involvement of downstream signaling molecules of CaSR in these effects.

## 2. Materials and Methods

All experiments were performed with the permission of the Institutional Animal Care and Use Committee of Nanjing Agricultural University (License number: SYXK-2017-0027).

### 2.1. Sampling and Perfusion of the Pig Duodenum

The proximal duodenums of three male (castrated) finishing pigs (Duroc × Landrace × Large White, weighing 110–130 kg) were collected from a local abattoir. The sampling procedure has been described previously [19]. Briefly, 4 cm tissue segments of the upper duodenum (at 4 cm after the pylorus) were obtained and immediately kept in oxygenized Krebs–Henseleit buffer (KHB) solution (pH = 7.2) containing 1.8 mM CaCl_2_ at 37 °C. Within 30 min, the duodenal tissue was transported to our laboratory for further processing.

Upon reaching the laboratory, tissues were washed thoroughly with 0.01 M PBS, and sliced into small pieces (approximately 1 mm^2^). Next, tissue samples were weighed and randomly placed into the chamber of the superfusion system (400 mg tissue/chamber) [20]. Before the start of the experiment, all the tissue samples were incubated in KHB with a constant flow of 6.00 mL/h for 40 min to allow them to reach an equilibrium state. Perfusion was performed for a total of 160 min. All perfusates were prepared using the KHB solution, as follows (Figure 1):

Experiment 1: To demonstrate the effect of Phe in the secretion of gut hormones, perfusion was performed using KHB solution (basic perfusate) supplemented with Phe (Sigma-Aldrich, St. Louis, MO, USA) at a final concentration of 0 mM (control group), 50 mM, or 80 mM. The experiment was repeated five times (five individual chambers for each treatment). Meanwhile, the tissues of the same batch were perfused in another three chambers for immunoblotting analysis.

Experiment 2: To determine the mechanism of CaSR contribution to gut hormone secretion stimulated by Phe in the porcine duodenum, perfusion was performed using KHB solution supplemented with 0 mM, 80 mM Phe, with or without 25 µM NPS 2143 (the CaSR antagonist), respectively. The experiment was repeated five times for each treatment. Meanwhile, the tissues of the same batch were perfused in another three chambers for immunoblotting analysis.

Experiment 3: Next, we evaluated whether Phe-induced CaSR activation required Ca^2+^, because [Ca^2+^]_e_ is a natural ligand of CaSR. This was determined by formulating a perfusate solution lacking in Phe and [Ca^2+^]_e_ (control group), or containing 80 mM Phe with or without 10 mM [Ca^2+^]_e_. The concentration (0–10 mM) of [Ca^2+^]_e_ was selected based on previous research [6,12]. The experiment was repeated five times for each treatment group.

For detecting CCK and GIP concentrations, perfusate samples were collected at 20 min intervals. For determining the mRNA expression of CaSR and genes encoding the gut hormones, the perfused tissue was obtained at the beginning and end of the perfusion, and then kept in liquid nitrogen until the pending analyses. For performing immunoblotting analysis, a small amount of tissue was excised (remaining tissue was put back in the chamber) after 10 min intervals, starting from 40 min to 110 min of perfusion in Experiment 1 and from the perfusion time of 50 min to 120 min in Experiment 2.

### 2.2. Total RNA Extraction, Protocol of cDNA Formation, and RT-qPCR

Extraction of total RNA from the perfused duodenum tissues of pigs was achieved using TRIzol regent (TaKaRa, Dalian, China) according to the recommended protocol. The concentration of extracted RNA was quantified using a NanoDrop ND-2000 spectrophotometer (Thermo Fisher Scientific, Waltham, MA, USA). For synthesis of cDNA, a total of 1 µg total RNA was used for reverse transcription using a reagent kit (PrimeScript™ RT, TaKaRa), following the recommended protocol. The synthesized cDNA was used to perform quantitative PCR (qPCR) by using the StepOnePlus™ Real-Time PCR System (Thermo Fisher Scientific) and SYBR^®^ Premix Ex Taq™ (TaKaRa, Dalian, China). The qPCR reaction is accordance with a previous report [18]. Each sample was assayed in triplicate. The list of primers used in the current study is given in Appendix A. The mRNA levels of CaSR was calculated as the ratio to GAPDH using the 2^−ΔΔCt^ method. The expression of CCK and GIP was calculated as the ratio to β-actin by employing the method of 2^−ΔΔCt^ for calculation.

### 2.3. Determination of Gut Hormone Concentration

CCK and GIP levels in the perfusate solutions were determined using corresponding enzyme-linked immunosorbent assay (ELISA) kits following the manufacturer’s protocols (Nanjing Angle Gene Biotechnology, Nanjing, China). The detection limits for CCK and GIP were 10 ng/L and 2 ng/L, respectively.

### 2.4. Western Blotting Analysis

The total proteins were extracted from the perfused duodenum tissues using lysis buffer (Thermo Fisher Scientific, MA, USA), and quantified using the bicinchoninic acid (BCA) method (Beyotime, Shanghai, China). Next, 50 µg of proteins were resolved by performing sodium dodecyl sulfate-polyacrylamide gel electrophoresis, and then transferred to polyvinylidene fluoride (PVDF) membranes (Millipore, MA, USA). The membranes were blocked with tris-buffered saline-Tween (TBST) containing 5% BSA for 1 h, and subsequently incubated overnight with primary antibodies: mouse anti-CaSR antibody (dilution, 1:2000; MA1-934; Thermo Fisher Scientific), anti-PKC antibody (dilution, 1:1000; ab23511; Abcam, Cambridge, MA, USA), anti-β-actin antibody (dilution, 1:1500; sc-47778; Santa Cruz Biotechnology), and anti-IP3R antibody (dilution, 1:1000; ab5804; Abcam). After washing four times with the TBST buffer, membrane incubation was done for 1 h with a goat anti-mouse IgG (H + L) secondary antibody (dilution, 1:5000; 31,160; Thermo Pierce, Thermo Fisher Scientific) or with a goat anti-rabbit IgG (H + L) secondary antibody (dilution, 1:5000; 31,210; Thermo Pierce, Thermo Fisher Scientific). Finally, the immunoreactive bands of protein were visualized using SuperSignal West Dura extended duration substrate (34,075; Thermo Pierce, Thermo Fisher Scientific), and then exposed to an X-ray film. Densitometric analysis of the protein bands was performed using BandScan 5.0 software (Glyko Inc., San Leandro, CA, USA).

### 2.5. Statistical Analysis

All data are expressed as mean ± standard error of the mean (SEM). Statistical analysis of the protein level between 0 mM vand 50 mM Phe in Experiment 1 was performed by an independent samples *t* test. Data of gut hormone concentration, mRNA expression, and protein level obtained from other experiments were statistically compared using one-way ANOVA, followed by the F-test for homogeneity of variance (*p* > 0.05) and the Student–Newman–Keuls test for multiple comparisons. All statistical analyses were performed using IBM SPSS Statistics software (version 20.0; IBM, Armonk, NY, USA), and the difference was set at *p* < 0.05.

## 3. Results

### 3.1. CaSR mRNA Expression and Gut Hormone Secretion after Treatment with Different Phe Concentrations

To determine whether Phe regulated CCK and GIP secretion, the mRNA expression of CaSR, CCK, and GIP and secretion of the two gut hormones were determined after treatment with the different Phe concentrations (Figure 2). CaSR expression was observed in the pig duodenum (Figure 2a). Compared to treatment with 0 mM Phe, treatment with 50 mM and 80 mM Phe significantly increased the relative mRNA expression of CaSR, CCK, and GIP (Figure 2a–c). Furthermore, treatment with 80 mM Phe sharply increased CCK concentration after 20 min of perfusion (Figure 2d) and GIP concentration from 80 min to 160 min of perfusion (Figure 2e). However, 50 mM Phe did not exert a similar stimulatory effect (Figure 2d,e). Thus, 80 mM Phe was employed in the subsequent experiments.

### 3.2. Phe-Regulated Expression and Secretion of Gut Hormones Was Mediated Via a CaSR-Dependent Mechanism

The CaSR inhibitor, NPS 2143 was employed to determine whether Phe-induced gut hormone secretion was dependent on CaSR. As compared to the control group, mRNA expression of CaSR, CCK, and GIP significantly increased with the addition of 80 mM Phe (Figure 3a–c). On the contrary, treatment with the CaSR antagonist NPS 2143 inhibited Phe-induced increase in CaSR, CCK, and GIP mRNA expression (Figure 3a–c). These findings indicate that Phe-induced gut hormone secretion is mediated by CaSR. Next, CCK and GIP secretion was analyzed when applied NPS 2143 to assess the potential role of CaSR (Figure 3d,e). Treatment with NPS 2143 suppressed the increased CCK and GIP levels induced by Phe (Figure 3d,e), suggesting that Phe-induced gut hormone secretion required CaSR.

### 3.3. [Ca^2+^]_e_ Modulated Phe-Induced CaSR and Gut Hormone Gene Expression and Gut Hormone Secretion

Figure 4 presents the summary for the role of Ca^2+^ in Phe-induced gut hormone secretion. Treatment with 10 mM Ca^2+^ and 80 mM Phe resulted in a significant increase in the mRNA expression of CaSR, CCK, and GIP in comparison to the control group (Figure 4a–c). However, no significant changes were observed in the mRNA expression of CaSR, CCK, and GIP after treatment with 80 mM Phe in the absence of Ca^2+^ (Figure 4a–c). Next, we evaluated the role of Ca^2+^ in Phe-induced gut hormone secretion. Employing treatment when 10 mM Ca^2+^ and 80 mM Phe were combined resulted in increased secretion of gut hormones (Figure 4d,e). However, treatment with 80 mM Phe when [Ca^2+^] was absent resulted only in increased CCK secretion (Figure 4d). Moreover, this increase was not obvious, compared with that observed after treatment with the combination of 10 mM Ca^2+^ and 80 mM Phe. These data indicate that Phe-induced gut hormone secretion requires Ca^2+^.

### 3.4. CaSR and Its Downstream Signaling Proteins Are Involved in Gut Hormone Secretion

Results of the western blotting analysis of CaSR and signaling proteins are shown in Figure 5. CaSR was also detected in the pig duodenum (Figure 5a). Compared with the control group, treatment with 50 mM Phe had no effect on the expression levels of CaSR and its downstream signaling proteins between 40 min and 100 min of perfusion (Figure 5a–d). However, treatment with 80 mM Phe significantly elevated CaSR, PKC, and IP_3_R levels from 40 min to 110 min of perfusion (Figure 5e–h). Moreover, CaSR and IP_3_R levels peaked after 80 min of perfusion. These results suggest that treatment with 80 mM Phe significantly increases the expression of CaSR and its downstream signaling proteins.

To further investigate the role of CaSR and its downstream signaling proteins in gut hormone secretion induced by Phe, we used the CaSR antagonist NPS 2143 to suppress CaSR activity (Figure 6a). Employing NPS 2143 significantly decreased Phe-induced CaSR expression during 50–70 min and 100–120 min of perfusion (Figure 6a,b). Moreover, treatment with NPS 2143 significantly decreased PKC and IP_3_R expression throughout the duration of perfusion (Figure 6a,c,d). These findings suggested that Phe induced gut hormone secretion by activating CaSR and its downstream signaling proteins.

## 4. Discussion

While a growing body of evidence has demonstrated that amino acids stimulate gut hormone secretion via multiple luminal sensing receptors in murine enteroendocrine cells, it is arbitrary to directly extrapolate these findings to the pig situation. Unfortunately, due to the high experiment costs and technical difficulties in performing research in vivo, fewer studies have uncovered the relationship between nutrition, sensors, and gut hormone secretion in pigs. Therefore, here we utilized in vitro porcine duodenal tissue to investigate the mechanism underlying Phe-induced CCK and GIP secretion, which is relatively easy and cheaper to conduct.

In the present study, we revealed that 80 mM, instead of 50 mM L-Phe, stimulated CCK and GIP secretion in the pig duodenum, although both of them increased the mRNA expression of CCK and GIP. This data is in line with our previous result in the pig stomach, where 80 mM L-Phe was required to trigger gastrin and somatostatin secretion [20]. Interestingly, our previous study suggested that 20 mM L-Trp was sufficient to induce CCK and GIP secretion under the same experimental conditions [18]. Other research disclosed that intraduodenally infusing a mixture containing amino acids such as valine, methionine, Trp, and Phe (47.2 mM) stimulated cholecystokinin (CCK) release, while the other mixture of amino acids, such as arginine, histidine, leucine/isoleucine, lysine, and threonine did not prove to exert the same effects [5]. These findings confirm that various structural characteristics and physicochemical properties of amino acids contribute to different secretory effects on hormone secretion. In addition, CCK concentration remained elevated in response to Phe during the 160 min perfusion period, which was comparable to that observed in published research conducted in the rat small intestine, lasting 90 min [6]. This may be explained in the context of the fixed rate at which Phe was constantly perfused, indicating that the pig duodenal tissue maintained sensitivity with Phe over the time. Furthermore, a rapid increase of CCK was observed after treatment with Phe, whereas the increase of GIP occurred until 80 min. It has been reported that CCK was secreted in response to amino acids and free fatty acids, while GIP secretion was mainly stimulated by luminal glucose [21,22]. Thus, we speculated that the different secretory response of CCK and GIP to Phe in our study may be associated with their individual sensitivity to luminal nutrients.

Recent studies have revealed that CaSR has not only been found in the thyroid gland, but also in the GIT to modulate gut hormone secretion [23,24]. Our qPCR and western blot findings confirmed that CaSR was indeed expressed in the pig duodenum. After treatment with 80 mM Phe, the expression of CaSR was markedly upregulated, demonstrating that there exists a link between Phe and CaSR. To further evaluate the role of CaSR in Phe-triggered CCK and GIP secretion, we perfused pig duodenum with different concentrations of [Ca^2+^]_e_ and the CaSR inhibitor NPS 2143. It is well-established that [Ca^2+^]_e_ is a natural agonist of CaSR and indispensable for gut hormone secretion induced by amino acids [25]. Our findings showed that in the absence of [Ca^2+^]_e_, 80 mM Phe failed to trigger the expression and secretion of CCK and GIP. Similarly, with the addition of NPS 2143, increased CCK and GIP secretion and their mRNA expression induced by 80 mM Phe were blunted. This concentration of NPS 2143 has been previously demonstrated to have no effect on cell viability and CCK secretion [26,27]. Therefore, these results collectively suggest that CaSR plays a vital role in Phe-induced gut hormone secretion.

CaSR-mediated signaling pathways have been studied for a few years. It was suggested that CaSR mediated gut hormone secretion, along with the elevation of [Ca^2+^]_i_ [28]. Nakajima et al. [29] reported that soybean β51–63 peptide induces CCK secretion through the activation of CaSR in enteroendocrine STC-1 cells, but this phenomenon disappeared after treatment with a [Ca^2+^]_i_ chelator. This finding indicated that the upregulation of [Ca^2+^]_i_ was indispensable for CaSR-mediated gut hormone secretion. IP_3_R and PKC are two vital CaSR downstream signal molecules, and both of them are involved in [Ca^2+^]_i_ oscillation [30]. However, whether IP_3_R and PKC participated in Phe-induced gut hormone secretion remained unclear. In the present study, 80 mM Phe rapidly increased the expression of PKC and IP_3_R at a protein level, whereas the expression was blunted by the CaSR antagonist NPS 2143, indicating that IP_3_R and PKC were responsible for Phe-induced gut hormone secretion mediated by CaSR. Therefore, we speculated that Phe induced gut hormone secretion through a CaSR-mediated signaling pathway involving PKC and IP_3_R activation. Certainly, to evaluate whether IP_3_R and PKC play direct roles in the regulation of gut hormone secretion, more evidence needs to be provided, such as inhibition of the activity or expression of IP_3_R and PKC.

## 5. Conclusions

In this paper, it was found that Phe evoked secretion of CCK and GIP in the pig duodenum through activation of the CaSR-triggered signaling pathway involving IP_3_R and PKC. The potential benefit of this study is the identification of the molecular mechanism of Phe-mediated anorectic effect, probing the impact of CaSR on appetite and feed intake in porcine production.

## Figures and Tables

**Figure 1 animals-09-00476-f001:**
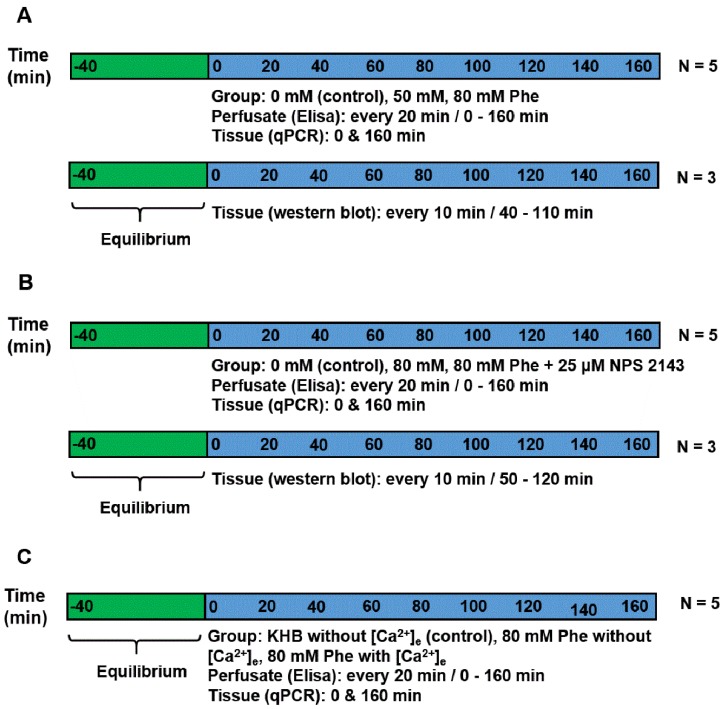
Experimental timelines. (**A**) Experiment 1: perfusion was performed using Krebs–Henseleit buffer (KHB) solution supplemented with Phe, having an ultimate concentration of 0 mM (control), 50 mM, or 80 mM. Perfusate was gathered at an interval of 20 min each to detect cholecystokinin (CCK) and glucose-dependent insulinotropic peptide (GIP) concentrations. Tissue was obtained at the beginning and end of the perfusion to check the mRNA expression. For immunoblotting analysis, the tissues of the same batch were perfused in another three chambers and collected from 40 min to 110 min. (**B**) Experiment 2: perfusion was performed using KHB solution supplemented with 0 (control group), 80 mM Phe, 80 mM Phe, and 25 µM NPS 2143. Perfusate was collected every 20 min to detect CCK and GIP concentrations. Tissue was obtained at the beginning and end of the perfusion to check the mRNA expression. For immunoblotting analysis, the tissues of the same batch were perfused in another three chambers and collected from 40 min to 110 min. (**C**) Experiment 3: perfusion was performed using KHB solution without [Ca^2+^]_e_, 80 mM Phe without [Ca^2+^]_e_, and 80 mM Phe with [Ca^2+^]_e_. Perfusate was gathered at intervals of 20 min each to detect CCK and GIP concentrations. Tissue was obtained at the beginning and end of the perfusion to check the mRNA expression.

**Figure 2 animals-09-00476-f002:**
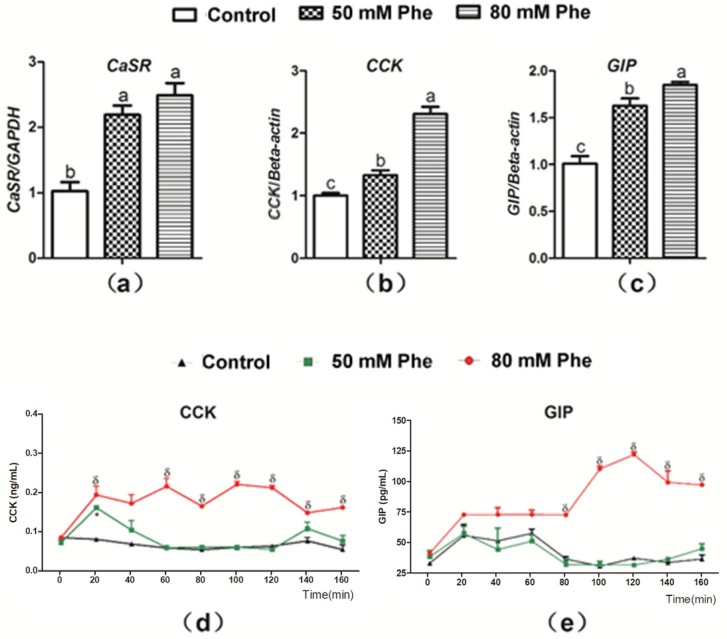
The mRNA expressions of CaSR, CCK, and GIP and the secretion of CCK and GIP in response to Phe in pig duodenum. The tissues were treated with different concentrations of Phe, namely 0 (control group), 50 mM, and 80 mM Phe. After 160 min of perfusion, the relative mRNA expression of (**a**) CaSR, (**b**) CCK, and (**c**) GIP was detected by Real-Time PCR; GAPDH and β-actin were used as internal controls. The concentrations of (**d**) CCK and (**e**) GIP were measured using corresponding ELISA kits every 20 min during the perfusion. Values are presented as mean and SEM (*n* = 5). For each gene (**a**–**c**), means with dissimilar letters show significant difference at *p* < 0.05. For the same time-points (**d**,**e**), significant differences at *p* < 0.05 are indicated by * and δ for Phe concentrations of 50 mM and 80 mM, respectively, compared with the control group.

**Figure 3 animals-09-00476-f003:**
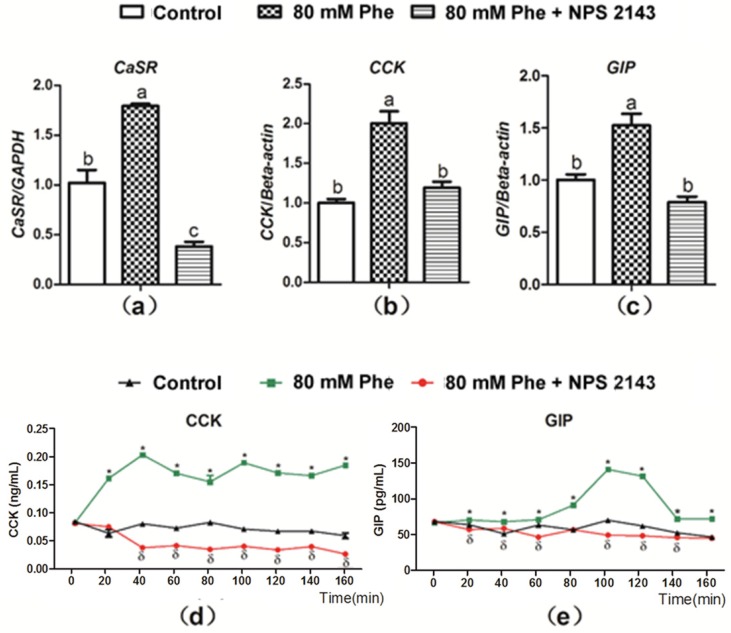
Effect of the CaSR antagonist NPS 2143 on Phe-induced mRNA expression of CaSR, CCK, and GIP, as well as secretion of CCK and GIP in pig duodenum. The tissues was perfused with 0 mM Phe (control group), 80 mM Phe, and 80 mM Phe with NPS 2143 (25 µM), respectively. After 160 min of perfusion, tissues perfusate were harvested to detect the mRNA expression of (**a**) CaSR, (**b**) CCK, and (**c**) GIP, where Real-Time PCR, GAPDH, and β-actin were used as internal controls. During the perfusion, perfusate solutions were collected at intervals of 20 min to detect the concentrations of (**d**) CCK and (**e**) GIP by the ELISA assay. Values are presented as mean and SEM (*n* = 5). For each gene (**a**–**c**), means without a common letter indicate a significant difference at *p* < 0.05. For the same time-points (**d**,**e**), significant differences at *p* < 0.05 are indicated by * and δ for treatment with 80 mM Phe and with the combination of NPS 2143 and 80 mM Phe, respectively, compared with the control group.

**Figure 4 animals-09-00476-f004:**
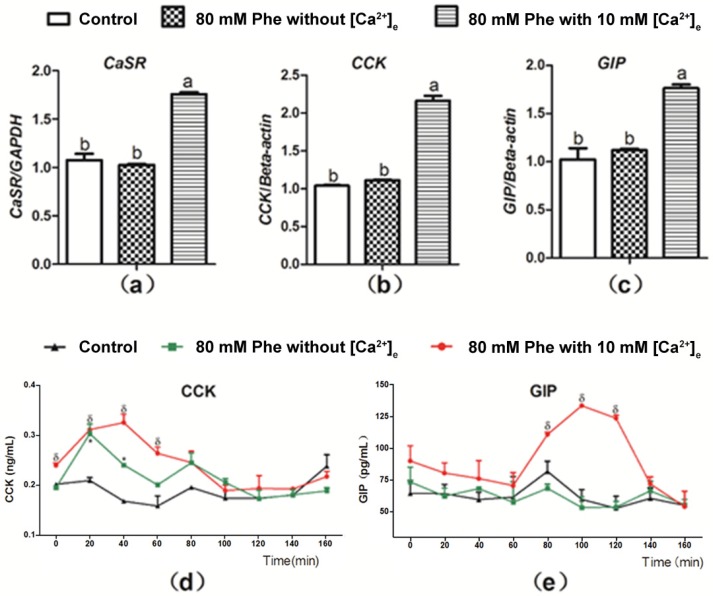
Effect of [Ca^2+^]_e_ on expression level of mRNA of CaSR, CCK, and GIP, as well as secretion of CCK and GIP induced by Phe in pig duodenum. Perfusion of duodenal tissues was performed with KHB without Phe and [Ca^2+^]_e_ (the control group), and 80 mM Phe with or without 10 mM [Ca^2+^]_e_. After 160 min of perfusion, the tissues were collected for qPCR analysis for expression of (**a**) CaSR, (**b**) CCK, and (**c**) GIP; GAPDH and β-actin were used as internal controls. The perfusate solutions were obtained at a 20 min interval to determine the concentrations of (**d**) CCK and (**e**) GIP using corresponding ELISA kits. Values are presented as mean and SEM (*n* = 5). For each gene (**a**–**c**), means with dissimilar letters differ significantly at *p* < 0.05. For the same time-points (**d**,**e**), significant differences at *p* < 0.05 are indicated using * and δ for treatment with 80 mM Phe in the absence of Ca^2+^ and in combination with 80 mM Phe and 10 mM Ca^2+^, respectively, compared with the control group.

**Figure 5 animals-09-00476-f005:**
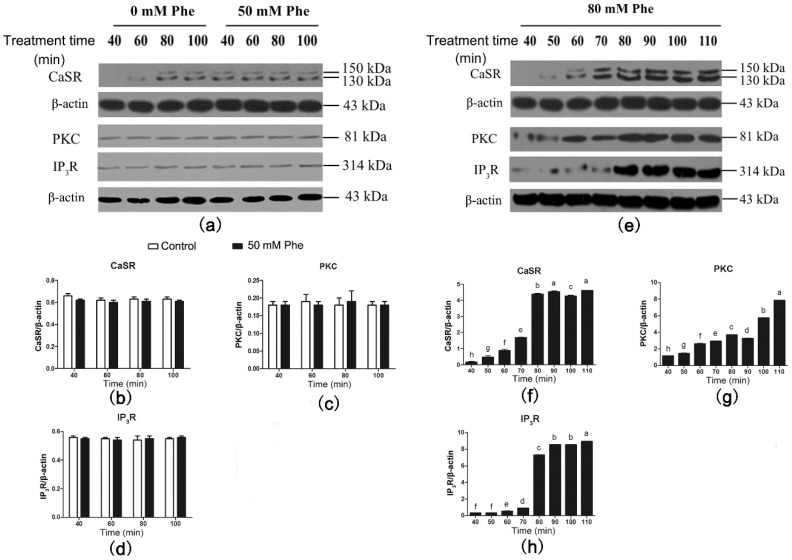
Effect of Phe (0 mM, 50 mM, and 80 mM) on the levels of CaSR-mediated downstream pathway in the pig duodenum. The perfusion of duodenal tissues were done with 0 (control group), 50 mM, and 80 mM Phe for 160 min, respectively. During the perfusion, the perfused tissues were recovered every 10 min, starting from 40 min of perfusion to 110 min of perfusion, to extract total protein. (**a**,**e**) The measurement of protein levels of (**b**,**f**) CaSR, (**c**,**g**) PKC, and (**d**,**h**) IP_3_R in the duodenum tissues was performed by western blot analysis, using β-actin as a loading control. Data are expressed as mean ± SEM (*n* = 3). Dissimilar letters represent a significant difference at *p* < 0.05.

**Figure 6 animals-09-00476-f006:**
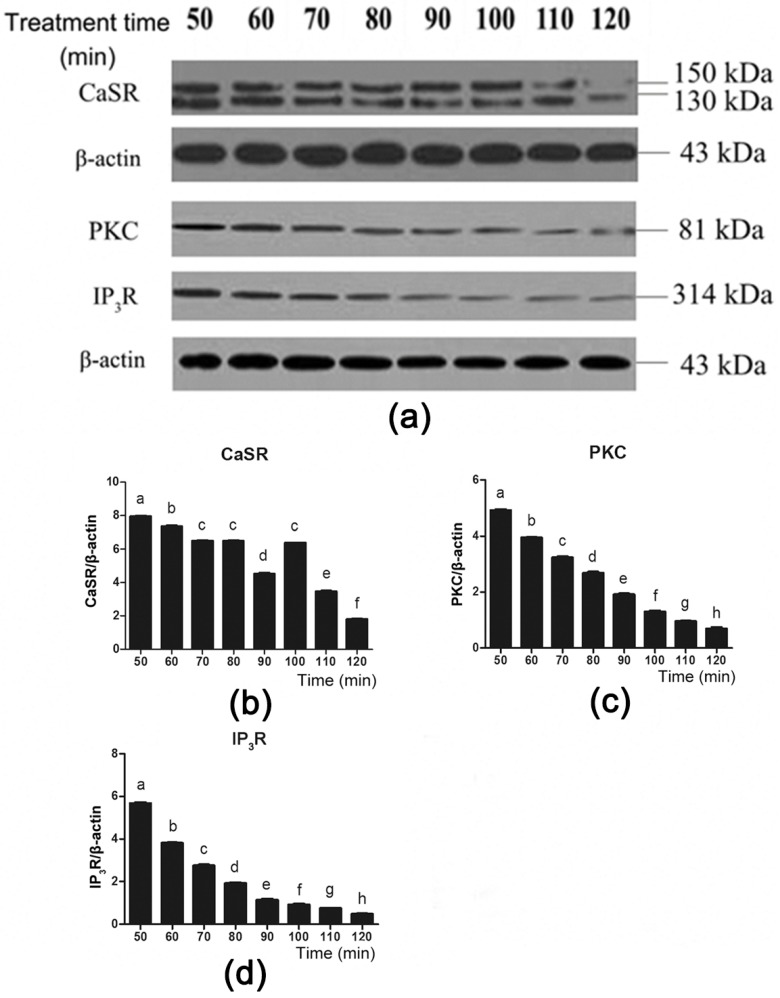
Effect of the CaSR antagonist NPS 2143 on the CaSR-mediated downstream pathway in the presence of 80 mM Phe in pig duodenum. The tissues were perfused with 80 mM Phe and 25 µM NPS 2143 (specific CaSR antagonist) for 160 min. During the perfusion, the perfused tissues were recovered every 10 min, starting from 50 min to 120 min, to extract total protein. (**a**) The measurement of protein levels of (**b**) CaSR, (**c**) PKC, and (**d**) IP_3_R in the duodenum tissues was performed by western blot analysis, using β-actin as a loading control. Data are expressed as mean ± SEM (*n* = 3). Dissimilar letters represent a significant difference at *p* < 0.05.

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
