# Peer review of "L-phenylalanine Increased Gut Hormone Secretion through Calcium-Sensing Receptor in the Porcine Duodenum"

_animals, 2019, doi:10.3390/ani9080476_

Round 1

Reviewer 1 Report

Manuscript ID: animals-505951

Title:
L-phenylalanine increased gut hormone secretion through calcium-sensing receptor in the porcine duodenum

Reviewers' comments:

Authors presented a paper related to how dietary factors affect the appetite in pigs in order to provide a strategy to modulate feed intake of pigs via L-phenylalanine (L-Phe) which has been demonstrated to induce satiety through stimulating anorectic hormone secretion in rodents.

The topic of the present study is very interesting and supported by multidisciplinary approach such as duodenal perfusion system, qPCR, ELISA, and Western blot analyses.

However, the article is not well presented, precisely in the material and method section. Introduction and discussion are quite clear and present the different aspects considered in the study, on the contrary material and methods are difficult to read and therefore the reader should continually go back on the section for reading.

Results are well descried and reflect what is presented in figures. In my opinion the big lack of the study is the very low number of animals used (samples from 3 pigs).

This study is a potentially good research, and manuscript can be considered for publication after major revision.

My specific comments can be found below. 

Major Points: 

Summary:

There’s a sentence related to obesity and its possible treatment, but the manuscript does not refer to pig as animal model for obesity, but only L-Phe for pig feed. Considering this, either the authors discuss the pig model in the manuscript or take it off the paper. Because the simple summary is the first part of the paper one can read, it’s deviant for the reader and it can create false expectations.

Materials and methods:

The timepoints are very complex to follow I suggest a table or a figure with type of analyses, substrate (medium or tissue) and timepoints.

Moreover, the number of animals used in the study is very low. I understand that you have performed many analyses at many timepoints but 3 animals seem to be exposed to bias linked to subjective variability.

Timepoint are very different for each analytical technique (qPCR at the end of the study, ELISA every 20 mins, WB every 10 mins starting from 50 or 60 minutes untile 110 or 120 mins): this experintal scheme is quite difficult to follow and there are no statements to specify the authors’ choice. After all, a correlation test for ELISA and WB should be very interesting if only the timepoints would be the same.

Did you study the power of the samples while designing the study?

In my opinion the statistical analyses should be performed with repeated measured model.

Minor points

Introduction:

·       Line 54-57: this sentence should be shifted in the discussion section.

·       Authors often refer to citation number 6 and 7 of the literature: in my opinion it’s better to specify the type to help the reader to better understand the topic.

·       Line 84-85: this sentence is a repetition of some previous ones.

·       Line 90 T1R1 in extenso

Material and methods:

·       Line 106: there’s no information about genetic ans sex of the animals

·       Line 111: duodenum was excised from the proximal part: can you please specify it considering the duodenal S curve?

·       Line 117: you decide for 160’ as endpoint of the study: please specify your choice

·       Line 183: Nor experimental Unit nor main factor are expressed, please specify.

Discussion

·       Lines 305-307. Not clear, please specify or refrase

·       Lines 315-316: . Authors state that “Our findings showed 80 mM Phe failed to trigger the expression and secretion of CCK and GIP in the absence of [Ca2+]e. Similarly, with addition of CaSR inhibitor NPS 2143, increased gut hormone secretion and their mRNA expression induced by 80 mM Phe were blunted.” But they don’t give any explanation about this behaviour (activation only at 50mM Phe).

Reviewer 2 Report

This is an interesting paper presenting interesting results.

My major concern is if the effect of individually added amino acids will be the same, if L-Phe is added as a part of amino acid mixture. The answer to this question would have an implication of the finding for pig industry.

It would be appropriate to present data showing if doses of L-Phe used in the in vitro experiments are achievable in pig nutrition

In discussion section data  on the effect of different dietary doses of L-Phe on pig would enhance the discussion

It is too early to transfer these data to public health issues

Author Response

Dear reviewer,

Thanks very much for your comments and suggestions about our manuscript submitted to Animals (Manuscript ID: animals-505951).Those comments are all valuable and helpful for revising and improve our manuscript. We have studied comments carefully and answered each of your points as below.If you have any questions about this paper, please don’t hesitate to let us know.

Sincerely yours,Jiangyin FengProf. Suqin Hang

Point 1: My major concern is if the effect of individually added amino acids will be the same, if L-Phe is added as a part of amino acid mixture. The answer to this question would have an implication of the finding for pig industry.

Response 1: Thanks for the reviewer’s very valuable question. Indeed, L-Phe exists in the gut or diet as a part of amino acid mixture, and investigating the secretory effect of mixed amino acids will provide a better guideline for pig industry. But, this kind of research was confronted with many challenges. For example, a previous report showed that intraduodenal infusion of a mixture of tryptophan (11 mM), valine (61.6 mM), methionine (38.2 mM) and phenylalanine (47.2 mM) stimulated cholecystokinin (CCK) release in healthy human (Colombel et al., 1988). Another study revealed that intraduodenal treatment with 49 mM L-Phe is sufficient to increased plasma CCK levels (Steinert et al., 2015). Although the latter demonstrated Phe is a strong stimulator of CCK secretion, it is still hard to determine whether the increased gut hormone in former report was induced by Phe or other amino acids. Currently, the association between gut hormones and nutrients in pigs is a new research field and just beginning. Various types of amino acids exert different effects on gut peptide secretion due to their specific structural characteristics and physicochemical properties. Thus, we think that identifying the role of individual amino acids in gut hormone secretion is the most important at this time. Certainly, we agree with the reviewer’s suggestion. With the deep recognition of the secretory effect of individual amino acids in pig intestine, it is necessary to investigate the hormone secretory response to mixed amino aicds, such as comparing the effect of amino acid mixture with or without Phe on gut hormone secretion in our future studies. It will be more meaningful for the development of pig industry.

List of references above description:

Colombel, J.F.; Sutton, A.; Chayvialle, J.A.; Modigliani, R. Cholecystokinin release and biliopancreatic secretion in response to selective perfusion of the duodenal loop with amino acids in man. Gut. 1988, 29, 1158-1166.

Steinert, R.E.; Landrock, M.F.; Horowitz, M.; Feinle-Bisset, C. Effects of Intraduodenal Infusions of L-phenylalanine and L-glutamine on Antropyloroduodenal Motility and Plasma Cholecystokinin in Healthy Men. J. Neurogastroenterol. Motil. 2015, 21, 404–413.

Point 2: It would be appropriate to present data showing if doses of L-Phe used in the in vitro experiments are achievable in pig nutrition.

Response 2: Thanks for the reviewer’s valuable comments. Unfortunately, previous publications tend to measure the plasma levels of Phe in fasting porcine and the concentration is 100 μM or so (Gao et al., 2018, Morales et al., 2012), while the levels of Phe in the pig duodenum after feeding have been rarely reported. One reasonable explanation for this may be the dynamic nature of the absorption and degradation of nutrients in the gut. However, based on our knowledge and experience, the concentration of Phe in porcine duodenum fed a standard diet is definitely hard to reach to 80 mM. A recent research demonstrated that 50 mM L-Phe stimulated CCK secretion in healthy men, but this concentration did not exert similar effect in our study. This phenomenon may be explained by species difference. Importantly, the reviewer’s valuable suggestion reminds us that we should detect the effects of a lower concentration of Phe such as 60 mM, 70 mM on gut hormone secretion.

List of references above description:

Morales, A.; García, H.; Araiza, A.; Htoo, J.K.; Cota, M.; Arce, N.; Cervantes, M. Effect of L-valine supplementation to a wheat-based diet with leucine excess on performance, gene expression, and serum concentration of amino acids. J. Anim. Sci. 2012, 90, 89–91.

Gao, K.; Pi, Y.; Mu, C.; Peng, Y.; Huang, Z.; Zhu, W. Antibiotics-induced modulation of large intestinal microbiota altered aromatic amino acid profile and expression of neurotransmitters in the hypothalamus of piglets. J. Neurochem. 2018, 146, 219-234.

Point 3: In discussion section data on the effect of different dietary doses of L-Phe on pig would enhance the discussion.

Response 3: Thanks for the reviewer’s valuable suggestion. According to the reviewer’s advice, we have consulted numerous publications related to Phe, gut hormone and satiety. Unfortunately, there are no feeding trials uncovering the effect of different dietary doses of L-Phe on gut hormone secretion and feed intake in pigs. General speaking, researchers prefer to investigate the secretory response of gut hormones to amino acids by gavage or intraduodenal treatment with individual amino acid rather than modulating the composition of amino acids in the diet. One reason for this phenomenon may be that the researchers worried the influence of other dietary nutrients will eliminate the effect of Phe itself. Actually, the reviewer’s good suggestion was consistent with our future experiment plan. It will be interesting to investigate the effect of chronic administration of high or low Phe diet on gut hormone secretion and satiety in pigs.

Point 4: It is too early to transfer these data to public health issues

Response 4: We agree with the reviewer’s good comment. Actually, another reviewer made a similar suggestion that it is not appropriate to draw the conclusion related with obesity. Taking the advice of two experts, therefore, “In addition, from a public health perspective, ingesting Phe-abundant food seems a more healthy eating habit in the prevention of obesity. It also offers a novel strategy to treat obesity through pharmacological therapy targeting the CaSR-mediated signal pathway” has been revised to “indicating that CaSR may be a potential target in the regulation of food intake in pigs.” This revision highlights our finding in pig industry. Please also see the details in the revision (Page 1, Line 23-24).

Round 2

Reviewer 1 Report

None